# DS-TG: Dynamical Systems as Accurate and Efficient Solvers for Time-Dependent Differential Equations

## Abstract

Time-Dependent Differential Equations (TDDEs) are central to modeling dynamic processes in various scientific and engineering systems. Numerical solvers typically provide reliable solutions but are burdened by prohibitive costs due to fine-grained discretization and iterative procedures. Recent machine learning (ML)-based approaches attempt to accelerate computation through fast ML inference, yet they are often trained on trajectories produced by numerical solvers, resulting in reduced accuracy and limited generalization. Designing a TDDE solver that achieves good accuracy and high efficiency remains a fundamental challenge. In this paper, we introduce DS-TG, a novel TDDE solver that employs Dynamical Systems (DS) as Trajectory Generators (TG), exploiting the intrinsic connection between DS and TDDEs. DS-TG leverages a DS-based processor whose physical states evolve continuously in real time according to carefully designed dynamics that directly emulate the target TDDE. This approach represents a novel paradigm fundamentally distinct from traditional discrete-time methods, offering inherent advantages in both accuracy and efficiency. Specifically, the continuous evolution of DS-TG can be seen as partitioning the target trajectory into a continuum of infinitesimal time steps, thereby reducing the problem to learning the trajectory gradient at each intermediate state of evolution. Building on this foundation, we further introduce two hardware-friendly techniques to enhance the dynamics design: (1) Laplacian-style interactions for effectively capturing spatial derivatives and (2) higher-order interactions for better representing higher-order temporal derivatives. Extensive experiments across representative TDDEs demonstrate that DS-TG achieves superior accuracy while delivering up to $10^4\times$ efficiency improvement compared to baseline methods.

## 1 Introduction

Time-dependent differential equations (TDDEs) form the mathematical backbone for modeling dynamic behavior across many scientific and engineering systems. From the heat equation governing thermal dynamics in advanced manufacturing systems (Foteinopoulos et al., 2018) to the wave equation describing electromagnetic propagation in next-generation communication networks (Jin, 2015), and reaction-diffusion equations capturing biochemical processes in living systems (Erban & Chapman, 2009), TDDEs are indispensable for understanding, predicting, and controlling time-evolving processes. Therefore, developing accurate and efficient TDDE solvers has been a persistent focus of research, driving applications in predictive simulations, real-time monitoring, digital twins, interactive design, and closed-loop control systems, where both accuracy and efficiency are critical.

Existing solutions range from conventional numerical solvers to ML-based approaches. Numerical solvers, such as finite difference and finite element methods, typically provide reliable solutions (LeVeque, 2002; Zienkiewicz & Taylor, 2005). However, achieving high accuracy usually requires fine-grained spatial and temporal discretizations coupled with extensive iterative procedures, which impose substantial computational and memory overheads that limit their potential for advanced scientific computing and many real-world applications. Recently, ML-based methods have emerged as promising alternatives for solving TDDEs (Raissi et al., 2019; Lu et al., 2021b; Gupta & Brandstetter, 2022). These methods employ sophisticated models trained on ground-truth data to replace

iterative procedures with fast ML inference, thus accelerating the solving process (Li et al., 2021; Brandstetter et al., 2022; Goswami et al., 2023). However, they generally rely on trajectories generated by numerical solvers and follow auto-regressive paradigms with fixed time steps to produce solutions. This reliance leads to reduced accuracy and limits generalization (e.g., being constrained to a fixed temporal resolution), thereby restricting their practical utility in many real-world applications. Therefore, there is an urgent need for novel approaches that can address these limitations and generate fast and accurate solutions.

Recognizing the fundamental connection between physical Dynamical Systems (DS) and TD-DEs, we propose that recently emerging DS-based processors (Afoakwa et al., 2021; Song et al., 2024b; Wu et al., 2025) represent a promising yet underexplored candidate for solving TD-DEs. Specifically, a DS consists of a particle or ensemble of particles whose states evolve continuously over time according to dynamics typically described by differential equations. DS-based processors are designed to physically embody such dynamical systems using CMOS electronic components, with their internal states naturally evolving under DE-governed dynamics. This continuous evolution enables real-time trajectory generation on a $\sim$1-watt DS-based processor, serving as a natural TDDE solver.

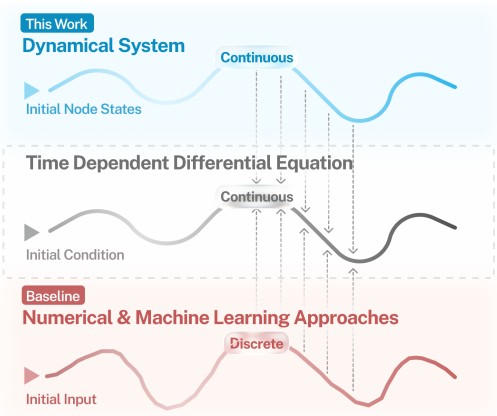

Figure 1: Overview of the proposed DS-TG.

Despite their compelling theoretical alignment with TDDE solving, the potential of current DS-based processors has only been demonstrated in limited domains such as optimization (Afoakwa et al., 2021; Sharma et al., 2023b; Sun et al., 2025) and graph learning (Song et al., 2024b; Wu et al., 2025), leaving their transformative potential for TDDE solving completely unexplored. In this work, we propose DS-TG, a DS-based solver for TDDEs that achieves good accuracy and revolutionary efficiency, initially realizing the potential of DS-based processors in TDDE solving. As depicted in Figure 1, DS-TG exploits the continuous evolution of DS-based processors to introduce a paradigm fundamentally different from traditional discrete-time methods, yielding a more natural approach to trajectory generation. Our key insight is that the continuous evolution of DS-based processors can be seen as partitioning solution trajectories into a continuum of infinitesimal time steps, thereby reducing the problem to learning the trajectory gradient at each intermediate state of evolution. Building on these learned gradients, DS-TG naturally evolves to produce the desired solution trajectories in real time on ultra-low-power DS-based processors. Instead of learning input-output mappings as in many ML approaches, DS-TG directly learns the underlying dynamics, which inherently provides better generalizability. To further enhance its modeling capability, we introduce two hardware-friendly techniques for its dynamics design. *(a) Laplacian-style Interactions:* We employ Laplacian-inspired node interactions to naturally capture spatial correlations and model the spatial derivative terms in TDDEs. *(b) Higher-order Interactions:* We incorporate higher-order interaction terms that enable accurate representation of higher-order temporal derivatives. These techniques collectively provide the expressivity required to represent diverse TDDE dynamics while maintaining hardware compatibility.

To the best of our knowledge, DS-TG represents the first attempt to leverage DS-based processors to solve TDDEs. The key contributions of this work are summarized as follows.

- We introduce DS-TG, a novel TDDE solver that leverages the intrinsic power of dynamical systems and their fundamental connection to TDDEs to accurately and efficiently solve a wide range of TDDEs across diverse domains.

- By exploiting the continuous evolution of DS-based processors, we reduce the TDDE solving problem as learning the trajectory gradient at each intermediate state of evolution, fundamentally departing from traditional discrete-time methods and enabling continuous solution generation.

- We propose hardware-friendly techniques including Laplacian-style interactions and higher-order interactions, enabling DS-TG to effectively capture diverse TDDEs.

- Experimental results on TDDEs across various scientific and engineering domains demonstrate that, compared to baselines, DS-TG achieves superior accuracy while delivering up to $10^4\times$ efficiency improvement.

## 2 PRELIMINARIES

This section introduces essential preliminaries for TDDEs and DS-based processors, including their mathematical foundations and architectures.

**Time-Dependent Differential Equations** form the mathematical foundation for modeling dynamic processes that evolve over time in physics, engineering, and applied sciences. A general representation of TDDEs can be written as:

$$F\left(t, x_1, x_2, \ldots, x_m, u, \frac{\partial u}{\partial t}, \frac{\partial u}{\partial x_1}, \ldots, \frac{\partial^2 u}{\partial x_1 \partial x_2}, \ldots\right) = 0, \tag{1}$$

where $t$ is the temporal variable, $x_1, x_2, \ldots, x_m$ are spatial variables, and $u$ denotes the unknown solution that depends on both time and space. Solutions are typically obtained under a combination of initial conditions (specifying the system state at $t = 0$) and boundary conditions (constraining spatial behavior). TDDEs are widely used in applications where capturing the system's temporal trajectory is essential, including weather prediction, plasma transport, and neural activity modeling.

**Dynamical System-Based Processors** mathematically embody a dynamical system that describes how components (nodes) interact and influence each other's states over time. Initially, DS-based processors are employed to address binary optimization problems (e.g., Max-Cut (Böhm et al., 2019; Liu et al., 2025c) and Satisfiability (Sharma et al., 2023a; Sun et al., 2025)) by minimizing a binary energy function (Hamiltonian), known as the Ising model. In addition to addressing optimization problems, DS-based processors have also been extended to a wide range of ML tasks characterized by real-valued Hamiltonian functions (Song et al., 2024b; Liu et al., 2025a;b), such as:

$$\mathcal{H}_{\text{RV}}(\mathbf{s}) = -\sum_{i \neq j}^{N} J_{ij}\sigma_i\sigma_j + \sum_{i}^{N} h_i\sigma_i^2. \tag{2}$$

where $\sigma_i \in \mathbb{R}$. $\mathbf{s} = \{\sigma_1, \sigma_2, ..., \sigma_N\}$ denotes the nodes in the dynamical system, $J_{ij}$ represents the relationship between node $\sigma_i$ and node $\sigma_j$, and $h_i$ refers to the self-reaction strength. The parameters $\mathbf{J}$ and $\mathbf{h}$ capture the relationship between system nodes.

The general architecture of DS-based processors is shown in Figure 2. Each node $\sigma_i$ is represented as a voltage on a capacitor $C$, while coupling parameters $\mathbf{J}$ and $\mathbf{h}$ are implemented as resistor conductance. To control and program the dynamical system, a set of Programming Units configures the parameters of the network (i.e., the resistance of the programmable resistors). The Column Select Unit manages column-wise programming of the couplers, while the Node Control Unit is in charge of node value initialization.

DS-based processors realize computation by allowing nodes to evolve continuously under well-designed dynamics, harnessing the natural flow of physical processes. For instance, the node dynamics can be designed as:

$$\frac{d\sigma_i}{dt} = \frac{1}{C}\left(\sum_{j \neq i}^{N} J_{ij}\sigma_j - 2h_i\sigma_i\right), \tag{3}$$

where $C$ denotes the capacitance of the circuit's capacitors. According to the hardware implementation, $\sigma_i$ is the voltage on a capacitor, while $J_{ij}$ and $h_i$ are resistor conductance. Consequently, the

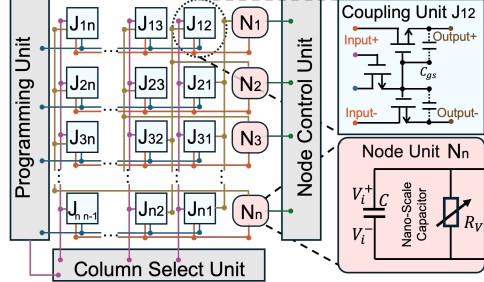

Figure 2: Architecture of DS processors.

terms $J_{ij}\sigma_j$ and $h_i\sigma_i$ correspond to electronic currents. Eq. 3 defines that the voltage of each capacitor $\sigma_i$ is continuously updated by the current $\sum_{j \neq i}^{N} J_{ij}\sigma_j - 2h_i\sigma_i$. Therefore, DS-based processors carry out computation through the charging and discharging of capacitor voltages, which operates at the "speed of electrons." By redefining the node dynamics, a DS-based processor can be reconfigured to efficiently and continuously execute a wide variety of computations.

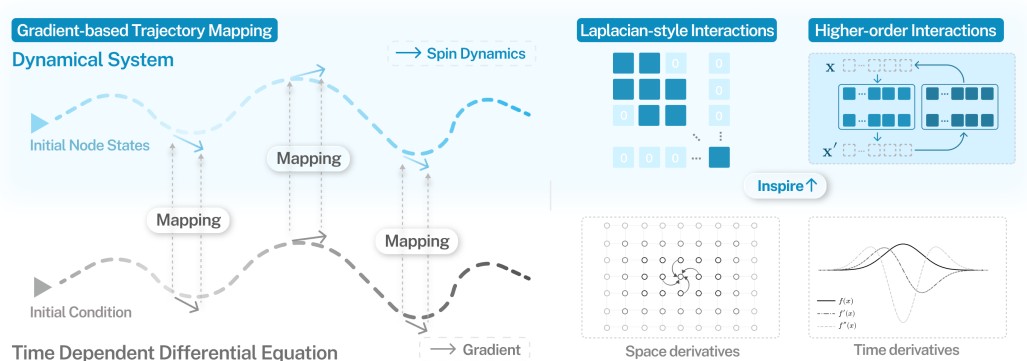

Figure 3: Workflow of the proposed DS-TG. The left panel illustrates Gradient-Based Trajectory Mapping, while the right panel depicts Physics-Inspired Dynamics Design.

## 3 METHODOLOGY

We present **DS-TG**, an accurate and efficient TDDE solver based on the continuous evolution of a CMOS-compatible dynamical system. As illustrated in Figure 3, DS-TG introduces two key innovations: (1) *Gradient-Based Trajectory Mapping*, which maps TDDE solving as the continuous state evolution on a DS, fundamentally eliminating the time discretization error; and (2) *Physics-Inspired Dynamics Design*, which instantiates DS-TG with physics-inspired dynamics to better represent the spatial and temporal derivatives in TDDEs.

### 3.1 GRADIENT-BASED TRAJECTORY MAPPING (GTM)

Traditional numerical methods approximate TDDE trajectories by discretizing time, which inevitably introduces discretization errors and often requires prohibitively small step sizes for stability, resulting in high computational costs. Recent ML approaches typically train on data generated by numerical solvers using autoregressive structures, where predictions are sequentially rolled out over a fixed horizon. These methods not only inherit the discretization errors of the underlying numerical solvers but also restrict generalization beyond the precomputed temporal windows.

To overcome these limitations, we propose GTM, a fundamentally different paradigm that establishes a direct mapping between the continuous-time gradient field of the TDDE with the intrinsic dynamics of a DS-based processor. Without loss of generality, a TDDE can be formulated as $\frac{d\mathbf{u}(t)}{dt} = f(\mathbf{u}(t))$, where $\mathbf{u}(t)$ denotes the system state and $f$ defines the gradient field governing its temporal evolution. A DS-based processor is itself a dynamical system whose physical states evolve according to:

$$\frac{d\mathbf{x}(t)}{dt} = g(\mathbf{x}(t); \theta), \tag{4}$$

where $\mathbf{x}(t) = [x_1(t), x_2(t), \ldots, x_n(t)]^T \in \mathbb{R}^n$ represents the processor nodes, $g$ denotes the parameterized dynamics, and $\theta$ represents the learnable parameters. The central idea of GTM is to align the TDDE gradient field $f$ with the processor dynamics $g$. This alignment is established by discretizing the spatial domain of the TDDE into grids and mapping them onto the processor nodes. We then design and optimize $g$ such that $g(\mathbf{x}(t)) = f(\mathbf{u}(t))$. Under this mapping, the physical trajectory of the processor state $\mathbf{x}(t)$ naturally emulates the solution trajectory $\mathbf{u}(t)$ of the original TDDE. Since the DS-based processor evolves continuously at the "speed of electrons" and with ultra-low power consumption, the solving process is both continuous and highly efficient. Essentially, the proposed GTM offers advantages:

- *Continuous Generation.* Unlike discrete-time schemes, GTM leverages the natural evolution of a DS-based processor to generate trajectories in continuous time, thereby eliminating the time discretization error.
- *Robust Generalization.* Instead of learning input–output mappings as in many ML approaches, GTM encodes the underlying dynamical laws, ensuring robust generalization.

- *Extraordinary Efficiency.* By transforming TDDE solving into natural evolution on a DS-based processor, GTM enables efficient trajectory generation at the "speed of electrons".

## 3.2 Physics-Inspired Dynamics Design

Through GTM, TDDE solving is effectively mapped to the continuous evolution of a DS-based processor. The continuous generation can be viewed as partitioning the trajectory into a continuum of infinitesimal time steps. The infinite steps make the required update at each step correspondingly simple, enabling us to design dynamics that are both expressive enough to capture complex TDDE structures and lightweight enough for CMOS hardware compatibility.

To ensure compatibility with the resistor-capacitor architecture of DS-based processors, we begin with the base dynamics as:

$$\frac{dx_i}{dt} = \sum_{j=1}^{n} W_{ij} x_j + b_i, \tag{5}$$

where $x_i$ represents the $i$-th processor node, $W_{ij}$ is the coupling weight encoding the interaction between node $x_i$ and node $x_j$, and $b_i$ represents external driving sources. To better capture TDDE dynamics, we enrich the basic dynamics with two physics-inspired mechanisms: (1) Laplacian-style interactions for better capturing spatial derivatives and (2) higher-order interactions for better representing higher-order time derivatives.

**Laplacian-Style Interactions for Spatial Derivatives.** TDDEs typically contain spatial derivative terms, such as the Laplacian operator $\nabla^2 u = \frac{\partial^2 u}{\partial x_1^2} + \frac{\partial^2 u}{\partial x_2^2} + \cdots$. To effectively capture these spatial derivatives, we introduce Laplacian-style interactions. Consider a spatial domain discretized into a regular $n$-dimensional grid where each grid point corresponds to a processor node $x_i$. The discrete Laplacian at node $i$ can be approximated using finite differences:

$$\nabla^2 u_i \approx \sum_{j \in \mathcal{N}(i)} w_{ij}(u_j - u_i), \tag{6}$$

where $\mathcal{N}(i)$ denotes the set of neighboring nodes and $w_{ij}$ represents the finite difference weight. Drawing inspiration from this mathematical structure, we further incorporate Laplacian-style interactions in the dynamics of DS-TG:

$$\frac{dx_i}{dt} = \sum_{j=1}^{n} W_{ij} x_j + \sum_{j \in \mathcal{N}(i)} L_{ij} x_j + b_i, \tag{7}$$

where $\mathbf{L}$ is a sparse coupling matrix with trainable parameters $L_{ij}$ for neighboring node pairs $j \in \mathcal{N}(i)$ and zero elsewhere, following the connectivity structure of the discrete Laplacian operator. This formulation naturally preserves the local connectivity patterns of spatial derivatives, while its sparse structure also ensures efficient hardware implementation.

**Higher-order Interactions for Temporal Derivatives.** While the above design effectively captures spatial interactions encoded by the spatial derivatives of TDDEs, extending DS-TG to handle higher-order temporal derivatives requires additional techniques. We introduce higher-order interaction terms that enrich its ability to represent more complex temporal structures. The key insight is to augment the processor state space to include auxiliary nodes that track higher-order temporal information. Specifically, for a TDDE involving up to $k$-th order temporal derivatives, we expand the processor state from $\mathbf{x} = [x_1, x_2, \cdots x_n]^T$ to an augmented hierarchical state representation $\tilde{\mathbf{x}} = \left[\mathbf{x}^{(0)}, \mathbf{x}^{(1)}, \cdots \mathbf{x}^{(k-1)}\right]^T$, where $\mathbf{x}^{(m)} \in \mathbb{R}^n$ represents the $m$-th order temporal derivative information, with $\mathbf{x}^{(0)} = \mathbf{x}$ corresponding to the original state nodes. This augmented state space enables us to construct a coupled dynamical system that naturally preserves the hierarchical structure of temporal derivatives. The governing dynamics for this augmented system are designed as:

$$\frac{dx_i^{(m)}}{dt} = \sum_{j=1}^{n} H_{ij} x_j^{(m+1)} \quad \text{for} \quad m = 0, 1, \cdots, k-2, \tag{8}$$

where $H_{ij}$ is a learnable coupling matrix that encodes the interaction strength between different state components across temporal derivative orders. This hierarchical coupling scheme maintains mathematical consistency with the TDDE formulation and enables efficient gradient propagation across multiple temporal scales.

### 3.3 HARDWARE IMPLEMENTATION

The overall dynamics design maintains compatibility with the resistor-capacitor (RC) architecture of DS-based processors, enabling direct mapping from mathematical formulation to analog circuit implementation. Specifically, each node $x_i^{(m)}$ is physically embodied as the voltage across a capacitor. The learned coupling matrices $\mathbf{W}$, $\mathbf{L}$, and $\mathbf{H}$ are realized through programmable resistive coupling units, consistent with those employed in previous DS-based processors. Each matrix element is mapped to the conductance of a programmable resistor, with the conductance directly encoding the corresponding coupling strength. The vector $\mathbf{b}$ denotes additional current sources integrated into the system. As a result, the weighted interactions $\sum_{j=1}^{n} W_{ij}x_j + \sum_{j \in \mathcal{N}(i)} L_{ij}x_j + b_i$ and $\sum_{j=1}^{n} H_{ij}x_j^{(m+1)}$ are naturally realized as currents flowing through the resistive coupling network and accumulating on the capacitors. These currents charge and discharge the capacitor voltages, implementing the continuous-time update dynamics in hardware. The resulting implementation faithfully reproduces the designed dynamical system while harnessing the computational efficiency and inherent parallelism of RC networks, making it particularly suitable for real-time processing applications requiring low power consumption and high throughput.

## 4 EVALUATION

### 4.1 EXPERIMENTAL SETUP

**Datasets.** We conduct experiments on five widely-evaluated TDDEs, including Heat equation, Advection equation, Wave equation, Burger equation, and Reaction-Diffusion (R-Diffusion) equation. These benchmark equations capture a broad spectrum of physical phenomena, ranging from diffusion and transport to nonlinear convection, oscillatory dynamics, and spatiotemporal pattern formation. Detailed mathematical formulations, initial conditions, and boundary conditions for each TDDE are provided in the Appendix A.2.

**Baselines.** Following Takamoto et al. (2022), we compare our approach against representative baselines spanning different methodological paradigms: (1) *Physics-Informed Neural Networks (PINNs)* (Raissi et al., 2019): neural networks that embed governing physical laws into the loss function to guide training. (2) *Fourier Neural Operator (FNO)* (Li et al., 2021): operator learning framework that models solution mappings in the frequency domain. (3) *UNet* (Takamoto et al., 2022): convolutional encoder–decoder architecture adapted for solving TDDEs. Following standard practice, FNO and UNet are trained in the auto-regressive paradigm to generate trajectories with two temporal context regimes: (i) AR-1: single-step prediction using $u_t$ to predict $u_{t+1}$; and (ii) AR-16: multi-step context prediction using the previous 16 states $\{u_{t-15}, \ldots, u_t\}$ to predict $u_{t+1}$. Auto-regressive baselines use teacher forcing during training and free rollouts at evaluation for the full

Table 1: Accuracy comparison across baselines and TDDEs under ID conditions.

| Methods | **Heat** (ST) | **Advection** (ST) | **Wave** (ST) | **Burger** (ST) | **R-Diffusion** (ST) |
|---|---|---|---|---|---|
| PINN | 2.039e-2 | 9.746e-3 | 2.135e-4 | 1.324e-4 | 1.216e-3 |
| FNO (AR-1) | 2.807e-4 | 1.353e-4 | 3.752e-4 | 5.096e-3 | 1.522e-4 |
| FNO (AR-16) | 1.629e-5 | 2.699e-5 | 6.118e-5 | 2.500e-4 | 1.022e-4 |
| UNet (AR-1) | 9.341e-2 | 3.123e-4 | 1.702e-3 | 7.476e-3 | 8.681e-3 |
| UNet (AR-16) | 8.860e-2 | 2.577e-4 | 7.649e-4 | 6.823e-3 | 7.212e-3 |
| DS-TG | **1.052e-6** | **1.159e-6** | **1.343e-6** | **3.250e-5** | **1.606e-5** |

| Methods | **Heat** (LT) | **Advection** (LT) | **Wave** (LT) | **Burger** (LT) | **R-Diffusion** (LT) |
|---|---|---|---|---|---|
| PINN | 2.125e-2 | 9.963e-3 | 2.880e-4 | 1.976e-4 | 1.483e-3 |
| FNO (AR-1) | 7.076e-4 | 4.403e-3 | 1.619e-3 | 3.045e-2 | 1.835e-3 |
| FNO (AR-16) | 1.165e-4 | 1.561e-4 | 3.031e-4 | 5.219e-3 | 6.850e-4 |
| UNet (AR-1) | 1.368e-1 | 2.628e-2 | 1.823e-3 | 8.596e-3 | 4.432e-2 |
| UNet (AR-16) | 1.115e-1 | 1.651e-3 | 1.003e-3 | 7.047e-3 | 2.686e-2 |
| DS-TG | **1.149e-6** | **1.907e-5** | **1.591e-5** | **3.295e-5** | **2.724e-5** |

Table 2: Accuracy comparison across baselines and TDDEs under OOD conditions.

| Methods | Heat (ST) | Advection (ST) | Wave (ST) | Burger (ST) | R-Diffusion (ST) |
|---|---|---|---|---|---|
| PINN | 3.325e-2 | 9.815e-3 | 2.342e-4 | 2.864e-4 | 2.998e-3 |
| FNO (AR-1) | 1.443e-3 | 8.522e-3 | 3.924e-4 | 6.425e-3 | 3.123e-3 |
| FNO (AR-16) | 4.549e-5 | 5.740e-4 | 6.902e-5 | 5.791e-4 | 2.331e-3 |
| UNet (AR-1) | 1.426e-1 | 1.613e-2 | 3.848e-3 | 8.385e-3 | 3.176e-2 |
| UNet (AR-16) | 9.244e-2 | 6.975e-3 | 8.939e-4 | 7.284e-3 | 1.831e-2 |
| DS-TG | **1.356e-6** | **3.827e-5** | **7.450e-6** | **3.373e-5** | **5.761e-5** |

| Methods | Heat (LT) | Advection (LT) | Wave (LT) | Burger (LT) | R-Diffusion (LT) |
|---|---|---|---|---|---|
| PINN | 3.423e-2 | 1.047e-2 | 3.563e-4 | 2.991e-4 | 3.166e-3 |
| FNO (AR-1) | 8.113e-3 | 3.761e-2 | 5.810e-3 | 7.133e-2 | 7.465e-3 |
| FNO (AR-16) | 1.490e-3 | 1.290e-2 | 4.709e-4 | 6.811e-3 | 6.325e-3 |
| UNet (AR-1) | 1.613e-1 | 4.037e-2 | 1.164e-2 | 9.158e-3 | 4.968e-1 |
| UNet (AR-16) | 1.568e-1 | 2.454e-2 | 3.927e-3 | 8.445e-3 | 3.401e-1 |
| DS-TG | **1.808e-6** | **4.041e-5** | **1.617e-5** | **4.625e-5** | **6.564e-5** |

horizon. Detailed configurations and hyperparameter settings for all baseline methods are adopted from (Takamoto et al., 2022), ensuring consistency with prior work.

**Evaluation Protocol.** We systematically evaluate the proposed approach across two temporal scales to capture both short-term and long-term accuracy, as well as under distinct distributional settings to assess robustness and generalization capability. *Temporal Scales:* (i) Short-term (ST): free roll-outs over 100 time steps, measuring short-term prediction quality. (ii) Long-term (LT): free rollouts over 2000 time steps, evaluating stability and cumulative error over extended horizons. *Distribution Robustness:* (i) In-distribution (ID): test trajectories with initial conditions drawn from the same distribution as used in training. (ii) Out-of-distribution (OOD): test trajectories initialized with conditions outside the training distribution, assessing generalization capability.

**Experimental Platforms.** Experiments for PINN, FNO, and UNet are conducted on an NVIDIA A100 40GB SXM GPU, where we measure both accuracy and per-sample inference latency. For the proposed DS-TG, we build on the original hardware embodiment BRIM (Afoakwa et al., 2021), employing a custom CUDA-accelerated Finite Element Analysis (FEA) simulator to evaluate accuracy and latency. Accuracy is reported as the Mean Absolute Error (MAE), computed over the space–time rollout window. The power consumption of DS-TG is estimated using Cadence Mixed-Signal Design Environment with 180nm CMOS technology.

### 4.2 Accuracy Evaluations

**In-Distribution (ID) Performance Comparison** is presented in Table 1. DS-TG consistently out-performs all baselines across the five PDEs in both short-term (ST) and long-term (LT) regimes. The method achieves errors in the range of $10^{-6}$ to $4 \times 10^{-5}$, representing 1–3 orders of magnitude improvement over baselines. Long-term evaluations highlight DS-TG's exceptional stability: while most baselines suffer from severe error accumulation over 2000 time steps, DS-TG maintains remarkably consistent accuracy. Among baselines, PINNs attain moderate accuracy but lack the precision of operator-based models for complex dynamics. UNet architectures perform the worst, particularly in long-term scenarios. FNO exhibits stronger performance, especially with AR-16, but still lags behind DS-TG. Besides, the AR-16 vs. AR-1 comparison further shows that extended temporal context improves accuracy for baselines, though DS-TG achieves superior results regardless of context length.

**Out-of-Distribution (OOD) Performance Comparison.** Table 2 summarizes the OOD evaluation results, highlighting DS-TG's strong generalization capability. DS-TG maintains errors in the range of $10^{-6}$ to $7 \times 10^{-5}$ across all TDDEs and temporal horizons, showing only modest degradation from its ID performance. In contrast, baseline methods experience severe breakdowns under distribution shifts. For example, UNet (AR-1) fails catastrophically on the Reaction-Diffusion equa-

tion, with long-term errors escalating to $4.968 \times 10^{-1}$. FNO models also exhibit instability; for instance, FNO (AR-1) error on the Burgers equation grows from $6.425 \times 10^{-3}$ to $7.133 \times 10^{-2}$ under long-term OOD evaluation. Despite these challenging conditions, DS-TG consistently achieves 1–2 orders of magnitude better accuracy than the best-performing baselines, demonstrating that its learned dynamics generalize robustly beyond the training distribution.

Furthermore, Figure 4 presents temporal error evolution patterns via heatmap visualizations comparing different methods on the Wave equation under out-of-distribution (OOD) long-term conditions. PINN displays localized high-error regions while maintaining moderate overall performance. FNO variants (AR-1 and AR-16) exhibit varying degrees of error accumulation over time. The UNet approaches suffer from severe degradation, characterized by error propagation and substantial temporal instability. Most notably,

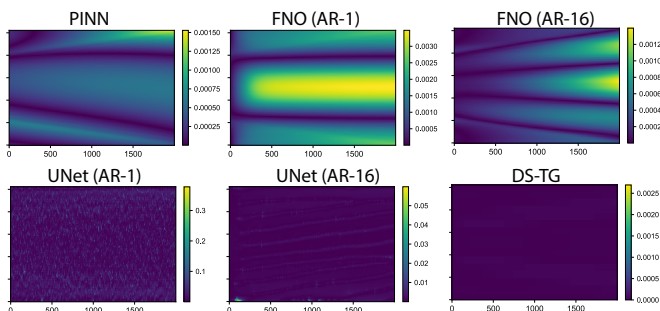

Figure 4: Temporal error evolution on Wave equation under OOD long-term conditions.

DS-TG maintains a remarkably uniform low-error distribution throughout the entire temporal domain. The consistently low error magnitudes demonstrate exceptional temporal stability and robust performance under distribution shifts. This visualization validates DS-TG's superior capability to preserve high accuracy across extended temporal horizons in OOD scenarios.

### 4.3 LATENCY AND POWER EVALUATIONS

Figure 5 presents the inference latency comparison across baselines and TDDEs, measured as the time required to produce one time frame. PINN demonstrates the lowest latency among the baselines, while both FNO and UNet operate at the millisecond level. Since DS-TG functions as a real-time con-

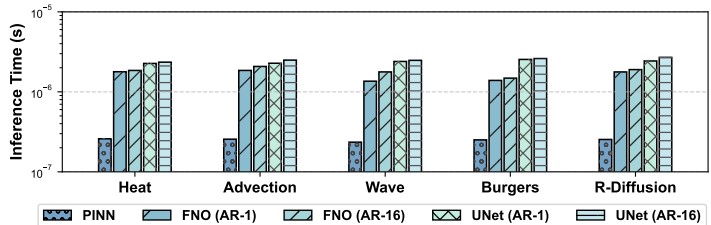

Figure 5: Solving latency comparison across baselines and TDDEs.

tinuous solver, unlike the discrete baselines, a direct latency comparison requires careful consideration. To establish a fair comparison, we evaluate the time required for all methods to evolve to a fixed TDDE state with equivalent accuracy. The baselines are trained on trajectories produced by numerical solvers and, once trained, operate with a fixed time step size $\Delta t$. This creates an inherent trade-off between time step size and accuracy: smaller $\Delta t$ values yield more fine-grained and accurate trajectories but require more computational steps. We therefore compare latencies when DS-TG and the baselines achieve the same accuracy level while evolving to a fixed system state. Our simulations demonstrate that DS-TG achieves a speedup of more than $10^4\times$ compared to the baseline methods. In terms of power consumption, DS-TG operates at approximately 132 mW in our experimental settings, while the A100 GPU typically consumes around 102W during operation. This represents a significant efficiency advantage, making DS-TG extraordinarily more efficient than the baseline approaches in both computational speed and power consumption.

## 5 RELATED WORK

**Solvers for TDDEs.** Numerical methods for solving TDDEs form the backbone of scientific simulation across diverse fields. They typically build on well-established discretization techniques, with the method-of-lines (Hamdi et al., 2007) representing a common strategy where the spatial domain

is discretized first using Finite Difference, Finite Element, or Finite Volume methods, reducing the TDDE to a large system of ordinary differential equations (ODEs) (LeVeque, 2002; Zienkiewicz & Taylor, 2005). The resulting ODE system is then integrated in time using explicit or implicit schemes. Explicit methods such as forward Euler and Runge-Kutta schemes (Akinsola, 2023) offer algorithmic simplicity and straightforward parallelization but are constrained by stability requirements that limit time step sizes, leading to prohibitively small time steps and correspondingly high computational costs (Hairer et al.). Implicit methods, including backward Euler and Crank-Nicolson schemes (Luskin et al., 1982), provide unconditional stability for many problems and allow larger time steps, but require solving large linear or nonlinear systems at each time step, introducing significant computational overhead and memory requirements (Ascher & Petzold, 1998).

Recent years have seen rapid growth in ML-based solvers for TDDEs. Physics-Informed Neural Networks (PINNs) approximate the solution with a neural network, and their carefully designed loss functions could enforce the underlying physical laws (Raissi et al., 2019; Karniadakis et al., 2021). This formulation allows PINNs to approximate the solution without explicit meshing. Despite their flexibility, PINNs often suffer from reduced accuracy and limited generalization across different initial or boundary conditions (Wang et al., 2021; Huang & Agarwal, 2023). In addition to PINNs, operator learning frameworks have also attracted significant attention, such as Fourier Neural Operators (FNOs) (Li et al., 2021), DeepONets (Lu et al., 2019; 2021a), and sequence-to-sequence surrogates (Brandstetter et al., 2022; Gupta et al., 2021). These methods aim to learn mappings from initial conditions and forcing functions to entire solution trajectories, providing better generalization across problem instances. Nevertheless, they usually exhibit degraded accuracy for long-term trajectory generation (Kovachki et al., 2023). Moreover, although ML-based solvers achieve faster inference than classical solvers, they still rely on digital processors and lack the ultra-low-latency response needed in real-time applications.

**DS-Based Processors** represent an emerging computational paradigm that has garnered substantial interest for their exceptional efficiency in solving optimization problems. A prominent example is the Ising machine, which physically implements the Ising model. These machines have demonstrated breakthrough performance across diverse NP-hard binary optimization problems, substantially outperforming conventional digital solvers. Applications span MAX-CUT (Haribara et al., 2016; Inagaki et al., 2016; Wang & Roychowdhury, 2019; Böhm et al., 2019; Mohseni et al., 2022; Grimaldi et al., 2023; Liu et al., 2025c; Ochs et al., 2021; Liu et al., 2025d; Cılasun et al., 2025), satisfiability (SAT) problems (Sharma et al., 2023a;b; Jagielski et al., 2023; Su et al., 2023; Bybee et al., 2023; Jin et al., 2025; Sun et al., 2025), and wireless communication (Singh et al., 2022; Sreedhara et al., 2023).

Recognizing their potential, DS-based processors have been extended to various machine learning applications, encompassing both binary and real-valued problem domains (Niazi et al., 2024; Wu et al., 2024; Song et al., 2024a;b; Liu et al., 2025a;b). However, existing approaches predominantly focus on the system's equilibrium state. They map ground truth to the system's equilibrium and obtain outputs by letting the system evolve until it converges to equilibrium. This paradigm neglects the rich information embedded within the system's evolution, a trajectory that naturally unfolds according to underlying dynamics, thereby failing to exploit a key computational advantage inherent to DS-based processors.

## 6 CONCLUSION

We introduced DS-TG, a dynamical-system-based solver that maps a target TDDE directly onto CMOS-compatible DS-based processors whose physical states evolve under differential-equation-governed dynamics. Leveraging the continuous-time evolution of DS-based processors, DS-TG transforms the TDDE solving problem as learning the trajectory gradient within each instantaneous state, thereby eliminating time discretization errors inherent in conventional approaches. Furthermore, two hardware-friendly techniques are introduced to improve the dynamics design of DS-TG: (1) Laplacian-style interactions for better capturing spatial derivatives, and (2) higher-order interactions for better representing higher-order temporal derivatives. Together, these techniques provide the expressivity required to represent diverse TDDE classes while maintaining hardware compatibility. Extensive experiments demonstrate that DS-TG delivers superior accuracy with orders-of-magnitude efficiency gains ($\sim 10^4\times$) over ML-based solvers across representative benchmarks.

**Reproducibility Statement.** To facilitate reproducibility, we provide comprehensive experimental details in Section 4.1 and in Appendix A.2. These resources can help independent researchers reproduce our findings and build upon our work.

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

# A APPENDIX

## A.1 LLM USAGE

LLMs were used solely for language polishing in this work. All LLM-assisted text was reviewed, revised, and verified by the authors.

## A.2 EXPERIMENTS CONFIGURATION

In this section, we provide the detailed mathematical formulations, initial conditions, and boundary conditions used to generate the benchmark datasets for the five time-dependent differential equations (TDDEs) considered in this work. All equations are solved numerically using the method-of-lines approach, where spatial derivatives are discretized while time integration is performed using the `solve_ivp` function with the RK45 scheme from (Virtanen et al., 2020). Unless explicitly stated otherwise, periodic boundary conditions are applied across all spatial dimensions to ensure consistent treatment of domain boundaries.

**Two-Dimensional Heat Equation.** This equation models diffusive processes and serves as a fundamental example of parabolic partial differential equations. The governing equation is

$$u_t = \alpha \left( u_{xx} + u_{yy} \right), \qquad \alpha = 0.01, \tag{9}$$

where $u(x, y, t)$ represents the temperature field and $\alpha$ is the thermal diffusivity constant. This equation is solved on the unit square domain $[0, 1] \times [0, 1]$ using a uniform $16 \times 16$ spatial grid. The initial condition is a Gaussian temperature distribution centered at the domain midpoint:

$$u(x, y, 0) = \exp\left( -\frac{(x - 0.5)^2 + (y - 0.5)^2}{0.01} \right). \tag{10}$$

This configuration creates a localized heat source that subsequently diffuses throughout the domain. Periodic boundary conditions are imposed in both $x$ and $y$ directions, effectively creating a toroidal topology that eliminates edge effects.

**Two-Dimensional Advection Equation.** This equation represents the transport of a scalar quantity by a prescribed velocity field without diffusion or source terms. The mathematical formulation is

$$u_t + c_x u_x + c_y u_y = 0, \tag{11}$$

where the constant advection velocities are set to $(c_x, c_y) = (1.0, 0.5)$, creating a diagonal transport pattern across the domain. Similar to the heat equation, this problem is solved on the unit square $[0, 1] \times [0, 1]$ with a $16 \times 16$ grid resolution. The initial condition is prescribed as a linear ramp in the $x$-direction:

$$u(x, y, 0) = \frac{x}{10}, \tag{12}$$

which creates a gradient that is subsequently advected by the prescribed velocity field. Periodic boundary conditions ensure that material exiting one boundary re-enters from the opposite side, maintaining conservation properties.

**One-Dimensional Wave Equation.** The equation examines wave propagation phenomena through:

$$u_{tt} = c^2 u_{xx}, \qquad c = 1.0, \tag{13}$$

where $c$ represents the wave speed. To facilitate numerical integration using first-order time-stepping schemes, this second-order equation is reformulated as a coupled system of first-order equations:

$$\begin{aligned} u_t &= v, \\ v_t &= c^2 u_{xx}, \end{aligned} \tag{14}$$

where $v$ represents the time derivative of the displacement field $u$. The computational domain spans $[0, 1]$ with a refined spatial discretization of 256 grid points to adequately resolve wave propagation. The initial displacement is specified as a Gaussian pulse centered at the domain midpoint:

$$u(x, 0) = \exp\left(-\frac{(x - 0.5)^2}{100}\right), \tag{15}$$

while the initial velocity field is set to zero, $v(x, 0) = 0$. This configuration generates symmetric wave propagation in both directions from the initial disturbance. Periodic boundary conditions allow waves to wrap around the domain boundaries.

**One-Dimensional Burgers' Equation.** The equation combines nonlinear advection with diffusive effects and serves as a simplified model for fluid dynamics phenomena:

$$u_t + \frac{1}{2}(u^2)_x = \nu u_{xx}, \qquad \nu = 10^{-3}. \tag{16}$$

The small viscosity parameter $\nu = 10^{-3}$ creates a nearly inviscid flow regime where nonlinear steepening competes with weak diffusive smoothing. This equation is solved on the unit interval $[0, 1]$ using a fine spatial grid of 256 points to capture the development of steep gradients. The initial condition is chosen as a Gaussian profile positioned near the right boundary:

$$u(x, 0) = \exp\left(-\frac{(x - 1)^2}{100}\right). \tag{17}$$

This placement, combined with periodic boundary conditions, allows observation of shock formation and subsequent nonlinear evolution as the profile steepens due to the quadratic nonlinearity.

**Two-Dimensional Reaction–Diffusion Equation.** This equation incorporates both diffusive transport and local chemical reaction kinetics:

$$u_t = \gamma(u_{xx} + u_{yy}) + r\, u(1 - u), \qquad \gamma = 0.01, \; r = 5.0. \tag{18}$$

Here, $\gamma$ controls the diffusion rate while $r$ governs the strength of the logistic reaction term $u(1-u)$, which exhibits bistable dynamics with stable states at $u = 0$ and $u = 1$. The equation is solved on the unit square domain $[0, 1] \times [0, 1]$ with a $16 \times 16$ spatial grid. The initial condition is prescribed as a linear gradient:

$$u(x, y, 0) = x, \tag{19}$$

creating a smooth transition from the unstable state $u = 0$ at the left boundary to the stable state $u = 1$ at the right boundary. This configuration promotes the formation of propagating reaction fronts that separate the two stable phases. Periodic boundary conditions are maintained in both spatial directions.

All solution trajectories are stored at 10,000 uniformly spaced time steps extending to the specified final time $t_{\text{final}}$ for each problem. The computational framework preserves both the solution field $u(\cdot, t)$ and the evaluated right-hand side function $f(u)$ at each time step, providing comprehensive data for subsequent analysis and validation of different approaches.

