# OpenReview forum: "DS-TG: Dynamical Systems as Accurate and Efficient Solvers for Time-Dependent Differential Equations"
_ICLR.cc/2026/Conference — ICLR 2026 Conference Withdrawn Submission_

### Official Review · Reviewer_hfLf · 2025-10-30

**Soundness:** 2
**Presentation:** 1
**Contribution:** 2
**Rating:** 4
**Confidence:** 3

**Summary:**

The paper introduces a novel approach for solving time-dependent differential equations by leveraging dynamical systems as trajectory generators. The authors claim that this method achieves higher accuracy and efficiency compared to baseline approaches.

**Strengths:**

The continuous-time mapping offers a highly appealing and insightful perspective, especially considering that most ML-based solvers operate in discrete time while physical systems evolve continuously.
In addition, the paper incorporates physics-aware design elements into the system, which may enhance its interpretability compared to conventional ML methods.

**Weaknesses:**

1. Presentation and Clarity: The overall presentation of the paper could be improved. Many important details are missing, making it difficult for readers without domain-specific knowledge to follow the content. For example, the workflows in Figures 1 and 3 are not sufficiently informative. I recommend revising the section “Dynamical System-Based Processors” to enhance clarity, and strengthening the connection between this background section and Section 3.2 to improve the logical flow.

2. Boundary Conditions: The paper does not provide sufficient discussion on how boundary conditions are handled, nor how different types of boundary conditions might affect the method’s implementation and performance.

3. Limitations and Implementation Details: A more in-depth discussion of the method’s limitations and implementation details would be valuable. For instance, how flexible is the proposed approach when applied to different systems? For more complex systems with irregular domains or diverse boundary conditions, does the architecture need to be redesigned for each case? How well does it generalize to high-dimensional systems? Finally, it would be helpful for the authors to elaborate on any observed failure modes of the model and potential solutions or mitigations.

**Questions:**

See comments above.

---

### Official Review · Reviewer_tbZs · 2025-10-31

**Soundness:** 3
**Presentation:** 2
**Contribution:** 2
**Rating:** 4
**Confidence:** 5

**Summary:**

The paper reframes TDDE solving via Gradient-Based Trajectory Mapping: it aligns the TDDE’s continuous-time gradient field with the intrinsic dynamics of a DS-based processor, effectively turning solution generation into learning the trajectory gradient along the evolution, thus enabling continuous (rather than stepwise) rollout.

**Strengths:**

(1)	By generating trajectories in continuous time on the processor, GTM removes time-discretization error and (by encoding governing dynamics rather than I/O mappings) supports robust generalization.
(2)	Strong accuracy and stability across PDEs.

**Weaknesses:**

(1) The manuscript’s description of TDDEs is not sufficiently precise. As presented, the hardware seems capable of emulating higher-order spatiotemporal interactions but does not appear to accommodate non-autonomous dynamical systems.
(2) The proposed hardware effectively functions as a continuous-time ODE simulator. Its incremental contribution and practical advantages over high-accuracy time-integration schemes are not clearly established.
(3) Unlike related work [1][2] that addresses questions central to the ML community, this submission focuses on differential-equation solving. The connection to machine learning and representation learning remains indirect.
(4) Closely related prior art [3] is not cited or discussed. Moreover, the venue targeted by [3] appears better aligned with the present paper’s scope and audience.

[1] DS-LLM: Leveraging dynamical systems to enhance both training and inference of large language
models. In The Thirteenth International Conference on Learning Representations, 2024a.
[2] DS-GL: Advancing graph learning via harnessing nature’s power within scalable dynamical systems. In 2024 ACM/IEEE 51st Annual International Symposium on Computer Architecture (ISCA), pp. 45–57, 2024b.
[3] DS-TIDE: Harnessing Dynamical Systems for Efficient Time-Independent Differential Equation Solving[C]//Proceedings of the 58th IEEE/ACM International Symposium on Microarchitecture. 2025: 1690-1703.

**Questions:**

(1) The manuscript does not clearly specify how boundary conditions are handled. How are they enforced in the implementation?
(2) Please clarify the pipeline from a TDDE specification to the hardware implementation. Does this process involve any learning, or is it a mapping?
(3) Does the hardware design exploit the explicit form of the governing equations, or does it rely solely on trajectory (state–time) data?
(4) Since the implementation depends on spatial discretization—and discretization choice affects accuracy—can you report experiments or ablations that quantify its impact?
(5) How does the approach perform on more complex nonlinear operators? For example, do you present results on the Navier–Stokes equations?
(6) Minor issue (L183). There is a grammatical error at line 183; please revise.

---

### Official Review · Reviewer_rtWe · 2025-10-31

**Soundness:** 1
**Presentation:** 1
**Contribution:** 1
**Rating:** 0
**Confidence:** 4

**Summary:**

This paper is concerned with developing a dynamical system (DS)-based processor framework that can solve time-dependent differential equations. The authors develop a DS-based processor compatible model with "Laplacian-style interactions" and "higher-order interactions for temporal derivatives", which ultimately leads to a high dimensional coupled ODE system with learnable parameters. Evaluating their model on different PDE systems, the authors posit that their method achieves much higher accuracy will using much less power.

**Strengths:**

1. Tuning an actual dynamical system onto the ODE / PDE of interest and using it to generate solution trajectories is a interesting idea. All the more so since the approach can potentially yield high accuracy solutions with low power consumption.

**Weaknesses:**

1. Overall, it is very unclear what exactly was implemented by the authors. The authors state that they build on the BRIM framework, but it is unclear if the authors actually did create a real-life hardware version of their model and ran their experiments on it, or if they just simulated the BRIM architecture using conventional machine learning libraries. The paper does not include any information on how the parameters of their DS-TG models were tuned, adding to the confusion.


2. The experiments seem inadequate for the problem setting considered. It seems like the paper is concerned with solving PDEs that are fully known in advance. However, the FNO and UNet models do not use this PDE information, and are just trained on some examples of the solution field (unknown equation problem). Therefore it is not surprising at all that FNO and UNet result in larger errors than the proposed method.
PINNs do use PDE information, and is the only baseline that seems relevant to the problem. However, PINNs are usually trained with much finer grid than the 16x16 grids considered in this paper, making it a very easy baseline to beat.
I believe, for the problem setting, a more adequate baseline is to use classical numerical solvers for ODEs / PDEs of lower order than the one used to generate the data.


3. I also feel that the "gradient-based trajectory mapping" lacks novelty. The idea of learning the vector field generating the time series data, instead of trying to model the solution values themselves is not entirely a new idea. Neural ordinary differential equations is an example of such a method, where the neural network is used to learn the unknown vector field from data.


4. The PDEs considered by the authors (as well as the resolution) are quite simple compared to the benchmark systems used in recent papers.

**Questions:**

In additions to the points raised in the Weaknesses section, here are some additional questions:

1. So what exactly was done? Did the authors actually create hardware implementation of their proposed framework, or did they just run numerical simulations? If the former, can the authors provide more detail on how the hardware was constructed?

2. The authors mention that the alignment between the PDE to solve and the processor dynamics is established by discretizing the original PDE into grids and mapping them on to the processor nodes. This seems to be the key learning algorithm. How was this done? Did the authors manually determine the parameters of the processor nodes? Is there an automated algorithm?

3. Regarding the experiments, how were each of the models trained? For PINN, did the authors just use 16x16 collocation points? For FNO and UNet, how were the training and test datasets constructed? For each model, how many parameters did each have?

4. Noting that derivative operations are also just linear operations, at the end of the day, the proposed model is just a high dimensional linear coupled differential equation with learnable coefficients. Is a linear model sufficient to model nonlinear phenomena? Can the authors explain why this linear model is able to predict the Burger's equation with good accuracy? What about other nonlinear equations?

5. Figure 5 does not contain the results for the authors' method. Please provide the relevant information.

6. Implementing this framework in the real world will introduce noise effects and other imperfections not present in the model formulation. Are such effects negligible or do the authors have a way to counteract those influences?

---

### Note · Authors · 2025-11-19

**Comment:**

We sincerely thank the reviewers for their time and constructive feedback. We will thoroughly address the raised concerns and suggestions, substantially improve the work, and submit a revised version to a future venue.

**Withdrawal Confirmation:**

I have read and agree with the venue's withdrawal policy on behalf of myself and my co-authors.